# Soluble Epoxide Hydrolase in Aged Female Mice and Human Explanted Hearts Following Ischemic Injury

**DOI:** 10.3390/ijms22041691

**Published:** 2021-02-08

**Authors:** K. Lockhart Jamieson, Ahmed M. Darwesh, Deanna K. Sosnowski, Hao Zhang, Saumya Shah, Pavel Zhabyeyev, Jun Yang, Bruce D. Hammock, Matthew L. Edin, Darryl C. Zeldin, Gavin Y. Oudit, Zamaneh Kassiri, John M. Seubert

**Affiliations:** 1Faculty of Pharmacy and Pharmaceutical Sciences, University of Alberta, Edmonton, AB T6G 2H7, Canada; kljamies@ualberta.ca (K.L.J.); darweshe@ualberta.ca (A.M.D.); dksosnow@ualberta.ca (D.K.S.); 2Department of Physiology, Faculty of Medicine and Dentistry, University of Alberta, Edmonton, AB T6G 2H7, Canada; hzhang10@ualberta.ca (H.Z.); saumya@ualberta.ca (S.S.); zhabyeye@ualberta.ca (P.Z.); kassiri@ualberta.ca (Z.K.); 3Department of Entomology and Nematology, University of California, Davis, CA 95616, USA; junyang@ucdavis.edu (J.Y.); bdhammock@ucdavis.edu (B.D.H.); 4National Institute of Environmental Health Sciences, NIH, Research Triangle Park, NC 27709, USA; matthew.edin@nih.gov (M.L.E.); zeldin@niehs.nih.gov (D.C.Z.); 5Department of Medicine, Faculty of Medicine and Dentistry, University of Alberta, Edmonton, AB T6G 2B7, Canada; oudit@ualberta.ca; 6Faculty of Medicine and Dentistry, Mazankowski Alberta Heart Institute, University of Alberta, Edmonton, AB T6G 2B7, Canada; 7Department of Pharmacology, Faculty of Medicine and Dentistry, University of Alberta, Edmonton, AB T6G 2H7, Canada

**Keywords:** soluble epoxide hydrolase, ischemic injury, failing heart, explanted hearts, sex differences, aging

## Abstract

Myocardial infarction (MI) accounts for a significant proportion of death and morbidity in aged individuals. The risk for MI in females increases as they enter the peri-menopausal period, generally occurring in middle-age. Cytochrome (CYP) 450 metabolizes N-3 and N-6 polyunsaturated fatty acids (PUFA) into numerous lipid mediators, oxylipids, which are further metabolised by soluble epoxide hydrolase (sEH), reducing their activity. The objective of this study was to characterize oxylipid metabolism in the left ventricle (LV) following ischemic injury in females. Human LV specimens were procured from female patients with ischemic cardiomyopathy (ICM) or non-failing controls (NFC). Female C57BL6 (WT) and sEH null mice averaging 13–16 months old underwent permanent occlusion of the left anterior descending coronary artery (LAD) to induce myocardial infarction. WT (wild type) mice received vehicle or sEH inhibitor, trans-4-[4-(3-adamantan-1-yl-ureido)-cyclohexyloxy]-benzoic acid (*t*AUCB), in their drinking water ad libitum for 28 days. Cardiac function was assessed using echocardiography and electrocardiogram. Protein expression was determined using immunoblotting, mitochondrial activity by spectrophotometry, and cardiac fibre respiration was measured using a Clark-type electrode. A full metabolite profile was determined by LC–MS/MS. sEH was significantly elevated in ischemic LV specimens from patients, associated with fundamental changes in oxylipid metabolite formation and significant decreases in mitochondrial enzymatic function. In mice, pre-treatment with *t*AUCB or genetic deletion of sEH significantly improved survival, preserved cardiac function, and maintained mitochondrial quality following MI in female mice. These data indicate that sEH may be a relevant pharmacologic target for women with MI. Although future studies are needed to determine the mechanisms, in this pilot study we suggest targeting sEH may be an effective strategy for reducing ischemic injury and mortality in middle-aged females.

## 1. Introduction

Ischemic heart disease (IHD) accounts for a significant portion of morbidity and mortality observed in individuals with cardiovascular disease (CVD). While mortality from both acute myocardial infarction (MI) and subsequent heart failure (HF) has declined, the clinical and financial burden remains significant [1]. Numerous risk factors such as smoking, obesity, and genetics contribute to adverse outcomes; however, the importance of age-related contributions to IHD are often overlooked [2,3]. The natural decline in cardiac function occurs progressively in the absence of comorbid medical conditions as individuals age. Our understanding of how age-related changes contribute to decreased function and reduced capacity to respond to stress remains limited.

Males and females demonstrate fundamental sex differences in cardiovascular aging that alter the lifetime risk for MI [4]. It is well characterized that women experience MI later in life than men [5]. However, while middle-aged, peri-menopausal women are less likely to have MI, they experience worse prognosis compared to their male peers [6,7]. Comorbidities or physician bias are unable to fully account for this difference, indicating an underlying risk for middle-aged women that remains to be identified. To effectively target and treat this population requires a comprehensive understanding of the underlying cardiac biology. The rate and progression of cellular aging can vary between individuals and species, but ultimately affects every cell in the organism. Accumulation of damaged mitochondria is an overarching mechanism associated with mammalian aging, which can increase an organ’s susceptibility to injury [8]. Intriguingly, studies have demonstrated that female rodent hearts contain less mitochondrial content, yet these mitochondria appear more efficient and more highly differentiated when compared to males [9]. Moreover, it is well characterized that mitochondria from female hearts demonstrate a greater degree of CV protection following ischemic injury, such as post-translational protein modifications, which result in decreased reactive oxygen species (ROS) generation and oxidative metabolism [10,11,12,13]. Yet despite these data, little is known about the interaction of age in female cardiac mitochondrial responses post-MI.

Linoleic acid (LA) and arachidonic acid (AA) are essential N-6 polyunsaturated fatty acids (PUFAs) released from cell membranes upon ischemic injury. LA and AA can be metabolised by cytochrome P450 (CYP450) epoxygenases into numerous active lipid mediators, termed oxylipids, which are readily converted into vicinal diols by the enzyme, soluble epoxide hydrolase (sEH) [14]. Previous studies demonstrated sEH inhibition or deletion has cardioprotective properties involving limiting mitochondrial damage caused by ischemic injury. Currently, no studies to date have focused on the effects of sEH inhibition in aged female hearts in the context of ischemic injury [15]. In this study, data from female explanted heart tissues demonstrated increased expression of sEH in patients with ischemic cardiomyopathy (ICM), correlating with damaged mitochondria. In this initial study we used an age-matched mouse model of ischemic injury to demonstrate that sEH inhibition elicits a cardioprotective effect in female mice, suggesting a potential therapeutic strategy to improve survival post-MI for middle-aged women.

## 2. Results

### 2.1. Clinical Parameters from Human Explanted Hearts

A summary of demographic and clinical parameters in female non-failing controls (NFC) and ICM human transplanted hearts is outlined in Appendix A

### 2.2. Epoxide Hydrolases Are Upregulated in Ischemic Human Left Ventricle (LV) Tissues

Expression of key epoxygenases and epoxide hydrolases was assessed by protein immunoblotting. CYP P450 isozymes, CYP2C8 and CYP2J2, are primarily responsible for LA and AA metabolism to the epoxyoctadecenoic acids (9,10- and 12,13-EpOME) and epoxyeicosatrienoic acids (5,6-, 8,9-, 11,12- and 14,15-EET) [14]. There were no significant changes in either CYP2J2 or CYP2C8 expression levels between NFC and ICM LV myocardium (Figure 1A,B). sEH is responsible for converting oxylipids, such as EpOMEs and EETs, into their vicinal diol forms, dihydroxy-9Z-octadecenoic acids (9,10- and 12,13-DiHOME) and dihydroxyeicosatrienoic acids (5,6-, 8,9-, 11,12- and 14,15-DHET), respectively [14]. sEH expression significantly increased in the non-infarct, peri-infarct, and infarct regions of female ICM patients, with the peri-infarct region demonstrating the highest increase in expression (Figure 1C). Due to its location tethered in the endoplasmic reticulum (ER), microsomal epoxide hydrolase (mEH) plays a role in basal epoxide hydrolysis, and can be upregulated following ischemic injury [16]. mEH expression in female hearts significantly increased in the peri-infarct and infarct regions (Figure 1D).

### 2.3. Human Left Ventricle Demonstrates Marked Changes in Oxylipid Metabolism Post-MI

Oxylipid levels quantified in female human and mouse LV tissue using LC–MS/MS are summarized in Appendix A. Assessment of human LV demonstrated significantly decreased amounts of AA throughout ICM hearts (Appendix A). Increased production of COX metabolites of AA (prostanoids), such as PGE_2_ and PGD_2_, were detected in non-infarct regions from human tissues, reflecting a pro-inflammatory response. CYP-derived epoxygenase metabolites of LA (12,13-EpOME and 9,10-EpOME) were increased in the non-infarct LV compared to NFC, and metabolites of AA (8,9-EET and 11,12-EET) were increased in the non-infarct region in females but not males (Appendix A). The epoxide hydrolase metabolites of the EpOMEs, 12,13-DiHOME, and 9,10-DiHOME significantly increased in the non-infarct LV (Appendix A). The ratio of 12,13-EpOME:DiHOME was significantly increased in the non-infarct region of female LV, an effect significantly attenuated in the peri and infarct regions (Figure 1E). The infarct region of female hearts also demonstrated a significantly decreased 9,10-EpOME:DiHOME ratio (Figure 1F). No changes in EET:DHET ratios was noted in human ICM hearts (Figure 1G–I). No levels of the ω-hydroxylase metabolites 19-HETE or 20-HETE were detected in the human LV. Hydroxyoctadecadienoic acids (HODEs) are lipoxygenase metabolites associated with oxidative stress. 13-HODE and 9-HODE, significantly increased in female ICM hearts (Appendix A).

### 2.4. Genetic Deletion and Pharmacologic Inhibition of sEH Preserves Survival of Mice Post-MI

We have reported that inhibition of sEH in vivo can protect against mitochondrial damage and cardiac dysfunction in young and aged mice, up to 7 days following LAD ligation [17,18]. Aged sEH null females demonstrated significantly prolonged lifespan compared to WT (Figure 2A). Serum *t*AUCB levels were significantly higher in female control and post-MI mice at 28 days compared to vehicle controls, confirming adequate *t*AUCB intake (Figure 2B), though total water consumption was comparable between all groups over the 28 days of treatment (Appendix A) [19].

To determine the chronic effects of permanent ligation in aged females, mice were followed for 28 days post-MI. Tetrazolium chloride (TTC) staining was used to ensure infarct in all groups (Appendix A). Female WT mice 28-days post-MI exhibited 58% survival, while sEH null mice demonstrated better survival rates at 73% (Figure 2C). Both *t*AUCB treatment groups demonstrated better survival at 80% for same-day treatment, and 83% for 4-day pre-treatment females (Figure 2C). Female CYP2J2-Tr mice subjected to LAD had a 100% mortality rate prior to the 28-day endpoint, which was attenuated by same-day administration of the sEHi *t*AUCB (Figure 2C).

Changes in body weight in females were negligible, as outlined in Appendix A. Wet lung weight normalized to tibial length (LW:TL) is a marker of pulmonary congestion following myocardial infarction, while heart weight normalized to tibial length (HW:TL) is a marker of hypertrophy. WT females demonstrated significantly increased LW:TL and HW:TL at 28-days compared to controls, an effect significantly attenuated in sEH null females (Figure 2D,E). Interestingly, female mice with same-day *t*AUCB treatment demonstrated significantly increased LW:TL and HW:TL, an effect not observed with *t*AUCB pre-treatment (Figure 2D,E). These data suggest that sEH genetic deletion protects the heart against maladaptive cardiac remodelling, correlating with improved survival. Similar effects were seen in pre-treatment, but not same-day, treatment with *t*AUCB, suggesting treatment timing may play a critical role in mediating sEHi response.

Cardiac sEH deletion was confirmed in sEH null mice (Figure 3A). In contrast to human hearts, WT mice demonstrated little change in sEH expression post-MI, while administration of *t*AUCB resulted in increased sEH expression (Figure 3A, Appendix A). Only the 4-day pre-treatment demonstrated increased sEH expression in the non-infarct region (Figure 3A). Female sEH null and *t*AUCB pre-treated mice had increased mEH expression in the peri-infarct regions (Figure 3B), suggesting a compensatory response to sEH inhibition in female mice.

Similarly to human tissues, female mouse hearts demonstrated increased production of prostanoids in non-infarct regions, suggesting a pro-inflammatory response (Appendix A). In contrast, the ω-hydroxylase metabolites, 19-HETE or 20-HETE, were not detected in human LV, while the non-infarct regions from sEH null and *t*AUCB same-day treated mice demonstrated lower levels compared to controls (Appendix A). Female sEH null mice had elevated levels of 17,18-EpETE, and also demonstrated a significant increase in 12,13-EpOME:DiHOME ratio, indicative of lower DiHOME formation in these hearts (Figure 3C,D). No such changes were observed in 9,10-EpOME:DiHOME ratios (Figure 3E). Similarly, 14,15-EET:DHET ratios also increased in sEH null female hearts (Figure 3F), while no changes were observed in other EET:DHET ratios (Figure 3G,H). Together, the shifts in oxylipid metabolism observed in sEH null females correlate with the preservations in survival and cardiac function observed in these mice.

### 2.5. sEH Genetic Deletion and tAUCB Pre-Treatment Protects Cardiac Function in Female Mice Post-MI

Cardiac function was assessed in all groups by conventional 2D echocardiography (ECHO) and electrocardiogram (ECG) at baseline, 7 days, and 28 days post-MI (Table 1). There were no differences in heart rate between groups that would account for the functional changes observed (Table 1, Appendix A). All groups of female mice demonstrated significantly reduced systolic function, %EF, and %FAC (Table 1). Conversely, sEH null and *t*AUCB 4-day pre-treated females demonstrated significantly preserved systolic function compared to WT at 7 days post-MI, suggesting an early degree of systolic preservation (Table 1). sEH null females also demonstrated preserved end-diastolic and end-systolic left ventricular chamber volumes (LVEDV;LVESV) at 28 days. Interestingly, both WT and sEH null females demonstrated prolonged E/E’ at 28 days (Table 1). In humans, E/E’ is a general marker of left ventricular diastolic pressure (LVDP) [20]. E/E’ was preserved in both groups of *t*AUCB treated mice at 7 and 28 days, suggesting preservation of diastolic function may be an important contributor to the survival effects observed with the sEHi treatment; however, general cardiac function appears more robust with pre-treatment of *t*AUCB (Table 1). *t*AUCB treatment in the absence of LAD occlusion did not have significant effects on heart function (Appendix A). A comparison of all cardiac functional and structural parameters in CYP2J2-Tr mice treated with *t*AUCB at baseline and 28 days post-MI is provided in Appendix A. Representative videos and m-mode images are included in the Appendix A.

ECG was used to assess changes in cardiac electrical signalling in control and post-MI mice, specifically QRS duration (ms) to assess ventricular action potential duration and PR interval for atrioventricular conduction time [21]. WT females exhibited increased QRS duration and PR interval, indicating an inability of the signals to propagate effectively to stimulate left ventricular contraction, effects attenuated in sEH null females (Table 1). QT prolongation, a hallmark of heart failure in both mice and humans [21], significantly increased in all groups at 7-days post-MI, but was better recovered in sEH null female mice at 28-days. The pattern of the QT prolongation confirmed the ubiquitous presence of HF, and matched the extent of systolic dysfunction observed in all groups (Table 1). Interestingly, females with same-day treatment of *t*AUCB demonstrated limited cardioprotection (Appendix A), and did not demonstrate the same significant shifts in oxylipid metabolism (Appendix A). Together, these data indicate that genetic deletion or pharmacological inhibition of sEH preserved post-MI systolic function and electrical conduction in female mice compared to WT counterparts. The effect of the pharmacological inhibitor on cardioprotection in females post-MI appears to be dependent on treatment timing.

### 2.6. Decreased Mitochondrial Function in LV Myocardium from ICM Patients

Defects in mitochondrial respiration are widely accepted as the driving force behind the cardiac dysfunction observed post-MI and throughout the progression to HF [22,23]. In order to assess function of the mitochondrial electron transport chain (ETC), we determined the enzymatic catalytic activity of ETC subunits: NADH:ubiquinone oxidoreductase (complex I), succinate dehydrogenase (SDH, complex II), and cytochrome c oxidase (COX IV, complex IV). Significant decreases in complex I and II activities were observed in peri-infarct and infarct regions of ICM female hearts (Figure 4A,B). Complex IV activity was significantly reduced in the non, peri, and infarct regions (Figure 4C). Citrate synthase (CS) activity, a marker of cellular aerobic metabolism due to its role as the rate-limiting factor for entry into the Kreb’s cycle [24], and an indirect marker of content, was significantly decreased in peri and infarct regions from female hearts (Figure 4D). Importantly, no changes in ETC protein expression were detected in any region assessed in ICM hearts from female patients (Appendix A).

### 2.7. Human LV Demonstrates Loss of Mitochondrial Ultrastructure and Organization

Mitochondria are dynamic organelles that undergo fission and fusion events in response to cellular stressors or metabolic requirements. Mitochondria undergo excess fission following ischemic injury, suggesting that preserving or promoting fusion processes may be beneficial [25]. Yet, there remain discrepancies in the literature regarding changes in fusion proteins in chronic post-ischemic cardiac remodelling. The process of mitochondrial fusion is driven by GTPAses, MFN-1, and MFN-2, which mediate outer mitochondrial membrane fusion, and optic atrophy 1 (OPA1), which mediates inner mitochondrial membrane fusion and preserves cristae structure [26]. Conversely, dynamin-related protein 1 (DRP1) translocates to the mitochondria from the cytosol initiating mitochondrial fission [27]. There were no significant changes in MFN-1, MFN-2, or OPA1 protein expression in mitochondrial fractions from human female explanted hearts (Figure 4E–G). DRP1 expression was decreased in both mitochondrial (Figure 4H) and cytosolic (Figure 4I) fractions in LV tissues from females. Electron micrographs of LV tissue demonstrated loss of mitochondrial ultrastructure and organization in regions from ICM hearts compared to NFC hearts in females (Figure 4J,K).

### 2.8. Female Mice with Genetic Deletion of sEH Demonstrate Improved Protein Expression and Mitochondrial Function Post-MI

Previously, we have shown declines in mitochondrial function without changes to mitochondrial protein expression of ETC enzymes in aged mouse hearts, suggesting an impact of overall quality [18]. Consistently, we observed no significant alterations in the expression of key proteins in mitochondrial oxidative metabolism, such as citrate synthase, complex I and II in hearts from WT, sEH null, or *t*AUCB treated mice (Figure 5A–D, Appendix A). However, citrate synthase activity was decreased in non- and peri-infarct regions of WT hearts post-MI, but not in sEH null mice (Figure 5E). Interestingly, significant increases in complex I activity were observed in post-MI hearts from female sEH null mice (Figure 5F). No significant differences in complex II or IV were observed in female mouse hearts (Figure 5G,H, Appendix A).

Assessment of LV mitochondria from female mice demonstrated increased OPA1 expression in non- and peri-infarct regions obtained from sEH null hearts compared to WT mice. Interestingly, in mice pre-treated with *t*AUCB for 4 days, all groups demonstrated increased OPA1 expression, including control (Figure 6A). MFN-1 expression in sEH null mice significantly increased in the non-infarct region of sEH null females compared to controls (Figure 6B). Similarly, mitochondrial MFN-2 expression significantly increased in the peri-infarct regions of sEH null and *t*AUCB pre-treated females compared to WT (Figure 6C). Mitochondrial DRP1 expression significantly decreased in the non-infarct region of sEH null females compared to controls, suggesting less mitochondrial translocation (Figure 6D). Cytosolic DRP-1 expression significantly declined in the non-infarct region of WT female mice, with no such change observed in sEH null or *t*AUCB pre-treated females (Figure 6E).

Analysis of mitochondrial O_2_ consumption was used to characterize mitochondrial function by determining respiratory rates such as basal state (resting or controlled respiration) and an ADP-stimulated state (active respiration and ATP synthesis), where the ratio represents a level of physiological efficiency expressed as respiratory control ratio (RCR). Interestingly, we observed a significant increase in RCR in the non-infarct region of WT females, while no change was observed in either sEH null or *t*AUCB pre-treated mice (Appendix A). However, there was a marked decline in corresponding ATP production from mitochondria in the post-MI WT hearts, which was increased in sEH null females and maintained in *t*AUCB treated mice (Figure 6F). Considering malate and glutamate were used as respiratory substrates, which are controlled exclusively by complex I, our observations of better complex I activity in post-MI hearts from female sEH null and *t*AUCB treated mice suggest that these hearts had preserved mitochondrial efficiency compared to WT female mice.

Sirtuins (SIRT1-SIRT7) are a class of nicotinamide adenine dinucleotide (NAD^+^)-dependent deacetylase proteins targeted therapeutically for ameliorating CVD [28]. In the heart, SIRT3 is mainly localized in the mitochondria where it is essential for preserving mitochondrial homeostasis. Importantly, SIRT3 regulates mitochondrial function by deacetylating, and thereby activating the different components of the ETC in addition to several mitochondrial proteins involved in modulating oxidative stress responses, energy metabolism, and mitochondrial dynamics [29]. In the current study, SIRT3 activity was significantly decreased in non-infarct regions of post-MI WT female hearts. However, SIRT3 activity was maintained in both sEH null mice or *t*AUCB pre-treated females subjected to MI (Figure 5I). Taken together, these data suggest preservation of mitochondrial SIRT3 activity is associated with conserved mitochondrial dynamic proteins and ETC complex activities, and thus improved overall survival in aged female mice with deletion of sEH or pre-treatment with *t*AUCB.

## 3. Discussion

In this study we characterized N-3 and N-6 PUFA metabolism in female explanted hearts obtained from NFC and ICM individuals, demonstrating key differences in sEH expression and metabolite profiles. Moreover, we used an age-matched animal model to demonstrate that aged female mice benefit from sEH inhibition or deletion following ischemic injury. The protective advantage includes preservation of both mitochondrial quality and cardiac function. Taken together, the data presented here suggest inhibiting sEH post-ischemia will have superior outcomes in females.

While animal models have demonstrated the role of sEH in mediating development and progression of CVD, data are limited regarding the role of altered CYP-derived oxylipid metabolism in humans [30]. Population analyses suggest genetic variation in the *EPHX2* gene is associated with the development of IHD [31], but sEH expression is tissue-dependent, and assessment in myocardial biopsies is relatively rare [30]. In this study, we first characterized myocardial specimens from patients with ischemic heart failure, and then utilized a mouse model of MI to demonstrate how changes in epoxide hydrolase expression and oxylipid metabolism correlate with cardiac and mitochondrial function. While sEH inhibition is cardioprotective in injury models, the importance of epoxide hydrolysis in removing endogenous and exogenous toxic metabolites highlights sEH as a mammalian clearance pathway [30]. Interestingly, inhibiting both sEH and mEH through genetic knock-out has been shown to be more effective than either alone in combating ischemic injury [16]. Future studies will need to confirm this in aged animal models.

Early research into the cardioprotective effects of sEH inhibition focused on production of the CYP-derived AA metabolites, EETs [32]. However, accumulating evidence suggests LA metabolites also contribute to adverse outcomes in IHD, which is supported by this study, wherein DiHOME levels were markedly increased in human myocardium obtained from ICM tissues. Previously, we demonstrated that a cardioprotective response observed in young mice with cardiomyocyte overexpression of CYP2J2 is lost in aged mice, which was attributed to increased DiHOME levels [33]. Recently, Bannehr et al. demonstrated perfusion with 12,13-EpOME and 12,13-DiHOME reduced post-ischemic cardiac recovery in young C57Bl/6 mice but co-perfusion of 12,13-EpOME with an sEH inhibitor protected the effect, suggesting the diol is toxic [34]. Interestingly, female CYP2J2-Tr mice treated with *t*AUCB following LAD ligation exhibited improved survival, supporting the notion that targeting sEH metabolism may be more beneficial than increasing epoxygenase activity alone.

The increased 12,13-EpOME:DiHOME and 14,15-EET:DHET ratios observed in sEH null female mice correlated with improved survival and cardioprotection, suggesting a metabolite profile including reduced DiHOMEs and increased EETs is important. Interestingly, treatment with an sEHi appeared to be highly time-dependent in our aged cohorts, as the mice pre-treated with *t*AUCB exhibited more substantial cardioprotection compared to the same-day sEHi treatment. These data suggest the timing of pharmacological inhibition has an effect on lipid availability and subsequent cardioprotection in this model of ischemic injury. While data regarding the co-regulatory effects of mEH and sEH over metabolite profiles involved in cardioprotective responses are limited, the increased expression of mEH detected in *t*AUCB-treated hearts potentially accounts for the reduced protection observed in these groups. Moreover, these results are consistent with an important metabolic role for epoxide hydrolases [16]. The timing and balance of different lipid mediators such as DiHOMEs and EETs can markedly influence how a heart responds to ischemic injury.

Mitochondria are fundamental facilitators of myocardial energy production and cell death pathways, as such, maintaining a healthy pool is essential to ensure proper heart function. Cellular quality control processes will adjust over a lifespan to accommodate cardiac growth and meet energetic needs [27]. Importantly, dysfunctional processes leading to decreased removal of damaged mitochondria or limited biogenesis of mitochondria are associated with adverse cardiac outcomes [35]. In the current study, no changes were observed in the expression of proteins involved in regulating mitochondrial dynamics from female ICM hearts [36]. Similarly, no differences were observed in age-matched WT mice despite these mice having significantly reduced cardiac function. Due to their critical role in inner mitochondrial membrane and cristae structure [35], the increased expression of mitochondrial dynamic proteins MFN-1/2 and OPA1 observed in sEH null or *t*AUCB treated female mice suggests better mitochondrial quality in these groups. These results are consistent with published data demonstrating sEH inhibition is associated with preserved mitochondrial quality in models of IR injury, and that oxylipids can regulate OPA1-dependent responses [37,38]. In contrast, diols like 12,13-DiHOME have been shown to decrease mitochondrial function in cardiomyocytes, potentially causing decreased cardiac function [39]. Interestingly, differences in ECG parameters, QRS duration, and PR interval observed in sEH null or *t*AUCB treated female mice support the notion of better calcium handling following LAD injury. Additionally, the increased levels of anti-arrhythmic 17,18-EEQ observed in female mice may contribute to these protective effects [40]. Importantly, mitochondrial preservation in sEH null or *t*AUCB treated female mice paralleled the improved survival rates, shifts in oxylipid metabolism, and attenuation of cardiac dysfunction, which were consistent with other studies of ischemic injury [18,38]. Taken together, these data support the hypothesis that alterations in the oxylipid profile, notably an increased availability of beneficial oxylipids and decreased level of detrimental diols, favours improved mitochondrial quality, leading to better post-ischemic cardiac function (Figure 6).

Accumulating literature has shown that one of the key functions of the mitochondrial SIRT3 is the deacetylation of ETC complex components, and consequently the promotion of effective electron transport via ETC to maintain ATP production and energy homeostasis [29]. Experimental studies have demonstrated that cardiac SIRT3 levels and/or activity decrease in response to MI [41]. Importantly, it has been well-documented that loss of SIRT3 is associated with increased acetylation of ETC complexes [42]. Increased acetylation of the ETC complexes correlates with a reduction in catalytic activity, impaired mitochondrial oxidative phosphorylation and consequently compromised contractile function of the heart. Therefore, approaches that upregulate and/or maintain SIRT3 expression and activity may represent a promising therapeutic strategy to treat or prevent cardiac dysfunction in the setting of MI [41,43,44]. In agreement with these reports, the current study demonstrated that the preserved SIRT3 activity in both sEH null mice or *t*AUCB pre-treated females subjected to MI correlates with enhanced cardiac contractile function compared to WT counterparts. Thus, indicating that inhibition of sEH is associated with maintained SIRT3 activity and amelioration of cardiac injury in response to MI.

While sex differences in sEH expression and activity have been reported in various tissues, including the heart, few studies have assessed these differences in aged models. We have recently demonstrated fundamental differences between sEH null males and females with natural aging, and that are associated with alterations in mitochondrial oxidative stress responses [15]. Moreover, evidence demonstrates sEH gene deletion provides protection in aged female mice following cerebral ischemia [45]. While no mechanisms have been identified, differences potentially involve epigenetic methylation of *Ephx2* by estrogen signalling, mediated through the estrogen receptors [46]. This study is the first to demonstrate robust responses to cardiac ischemia in aged female mice with sEH genetic deletion or pharmacologic inhibition. Differences in EET-mediated effects on the vasculature have also been reported, although the molecular mechanism remains unknown [47]. The effects observed in our current study suggest age-related changes that occur in females have a critical role.

There are several key limitations to the current study that require consideration and will need to be addressed in future research. A first limitation reflects the difficulty in obtaining age-matched NFC hearts to the ICM groups, which requires comparisons be made with caution. Importantly, female ICM patients in the current study encompassed an age range known to be peri-menopausal or menopausal in Canadian women [48]. Menopause presents at a wide range of ages, symptoms vary greatly between individuals, and the complete transition takes years to finalize [48,49]. Unfortunately, we were not able to ascertain the specific state of menopausal transition or assess the hormone levels in the current study, as such it was not possible to normalize these women to the peri-menopause, menopausal, or post-menopausal period. A comprehensive analysis of the menopausal transition would be an interesting and important future analysis to incorporate into a study. While mice and other rodents do not perfectly recapitulate the human menopausal state, without further surgical intervention, such as ovariectomy, the comparison to humans is limited in our study [49,50]. These analyses and determination of changes to the levels of sex steroid hormones in female mice will be essential in future research. In addition, natural aging processes occurring in humans are often confounded by numerous factors, including contraceptive use, pregnancy, and breast-feeding, limiting extrapolation of data from animal models [49]. Ultimately the observational nature of this study necessitates the need for future research to fully address the mechanisms involved, including the effects of sex hormones. Current on-going research into sexual dimorphism in our sEH null female and male mice is expected to clarify many of the limitations presented in this pilot study.

The data presented corroborate a growing body of work supporting the need for sex-specific research. Clinically, while middle-aged women are at a lower risk for MI compared to men, they demonstrate a higher risk for adverse events following ischemic injury including mortality [6]. The lack of data regarding age and sex interactions in ischemic injury leaves a “knowledge-gap” that needs to be addressed in pharmaceutical research.

In summary, we have demonstrated sEH genetic deletion or pharmacological inhibition slows the progression of cardiac and mitochondrial dysfunction in female mice following ischemic injury (Figure 7). Crucially, these effects were observed in aged mice, which better correlated to the age of humans with ICM in this study. Moreover, we demonstrated significant increases in sEH expression and altered oxylipid metabolite profiles in human left ventricular tissue from ICM explanted hearts, providing compelling evidence for targeting this pathway in humans. While future studies are essential to determine the mechanisms behind these effects, this preliminary study provides evidence that inhibiting sEH may be a promising therapeutic target for a vulnerable population.

## 4. Materials and Methods

### 4.1. Human Explanted Heart Tissue

Human heart tissues were procured under protocols approved by the Health Research Ethics Board of the University of Alberta. Adult non-failing control (NFC; N = 5 female) heart tissues (LVEF ≥ 60%) were collected from female (self-identifying) donors with no cardiovascular history when transplants were unsuitable due to medical or technical issues, such as ABO blood type incompatibility, as part of the Human Organ Procurement and Exchange (HOPE) at the University of Alberta. Adult failing heart samples were procured from female patients with end-stage heart failure secondary to ischemic cardiomyopathy (ICM; N = 5) as part of the Human Explanted Heart Program (HELP) program at the University of Alberta. Collections were conducted during cardiac transplantations at the Mazankowski Alberta Heart Institute (MAHI). Menopausal status was not ascertained at time of transplant, but all individuals meet the recognized age for peri-menopausal or menopausal state. All myocardium samples were excised from the left ventricular free wall avoiding epicardial adipose tissue, within 5–10 min of its excision following cold cardioplegia. Three samples from each ICM heart were obtained from the non-infarct (remote region, viable tissue), peri-infarct (border region containing viable and non-viable tissue), and infarct (direct injury) regions. The samples were immediately flash-frozen in liquid nitrogen and stored in ultra-low (−80 °C) freezers. Mitochondrial ultrastructure was assessed in left ventricular tissue from NFC and non-infarct regions from ICM hearts. Tissues were sectioned and examined by a transmission electron microscope, as previously described [18]. Detailed demographic and clinical information of all research subjects is summarized in Appendix A.

### 4.2. Animal Studies

Colonies of mice with targeted deletion of *Ephx2* (sEH null) or mice over-expressing cardiomyocyte-specific human cardiac epoxygenase CYP2J2 (CYP2J2-Tr) are maintained at the University of Alberta. Wild-type (WT) littermates from each colony served as controls. All colonies are maintained on C57Bl/6 background. Female mice 13–16 months old, roughly equivalent to “middle-age,” were used to age-match human tissues, and avoid confounding frailty effects of elderly mice. Hormonal status was not assessed in these females, but all mice were non-breeding. Mice with a baseline ejection fraction <48%, or an E:A ratio <1 as determined by echocardiography, were excluded from the study. All other mice were randomly allocated to pertinent control or experimental groups. WT mice were given sEH inhibitor trans-4-[4-(3-adamantan-1-yl-ureido)-cyclohexyloxy]-benzoic acid (*t*AUCB; 10mg/L) or vehicle (0.1% DMSO) in drinking water, either immediately following ligation (same-day treatment, *t*AUCB:0d), or 4 days prior to ligation (pre-treatment, *t*AUCB:4d-pre). Both groups received continued administration ad libitum in the drinking water for the duration of the experiment. Water intake was measured weekly when bottles were changed and fresh water administered. Animal procedures were carried out in strict adherence to the guidelines of Animal Care and Use Committee at the University of Alberta and the Canadian Council of Animal Care. These guidelines conform to the principles of the US National Institute of Health *Guide for the Care and Use of Laboratory Animals* (NIH Publication No. 85–23, revised 1985, Washingtion DC, USA).

### 4.3. Induction of Myocardial Infarction in Mice

To induce MI, aged female mice underwent permanent ligation of the left anterior descending coronary artery (LAD), as previously described [17]. Mice were anesthetized with 1–2% isoflurane delivered via nasal aspiration for 20–30 min. The surgeon was blinded to all genotypes and treatments. Following surgery, mice were administered an analgesic (meloxicam, 3–5 mg/kg q.d. to alleviate pain once per day for three consecutive days, and were allowed to recover for 28 days. Mice were 28 days post-MI and sacrificed by cardiac excision following lethal injection of sodium pentobarbital (100 mg/kg, i.p.) and tissues were collected. Upon necropsy, MI hearts were separated into non-infarct, peri-infarct, and infarct regions under a dissecting scope. Only non-infarct and peri-infarct areas were used for analysis due to limited tissue amounts.

### 4.4. Infarct Assessment

Following excision, hearts were rinsed in 1X PBS, frozen in liquid nitrogen and evenly transversely sliced from the apex to the point of ligation. Then, 1 mm slices were placed in a 1% solution of triphenyltetrazolium chloride (TTC), wrapped in tin foil and incubated in a tissue incubator at 37 °C for 15 min before removal and imaging. ImageJ software was used to calculate percent infarct taken as the ratio of the pale area (infarct) to total area [51].

### 4.5. Mouse Cardiac Function Assessment

Cardiac structure and function were assessed noninvasively in female mice by transthoracic 2D echocardiography 1 week prior to surgery (baseline), and then at 7 days and 28 days post-MI following anesthesia with 1–2% isoflurane. The Vevo 3100 high-resolution imaging system with 40MHz transducer (MX550S; Visual Sonics, Toronto, ON, Canada) was used for recording, with all analysis of acquired images done with Visual Sonics VevoLab software. Left ventricular internal diameters, as well as the diameters of the intraventricular septum (IVS) and posterior walls (LVPW), were taken from m-mode images acquired at mid-papillary level. Simpson’s modified method was used to determine systolic parameters, ejection fraction (%EF), fractional area change (%FAC), left ventricular end diastolic (LVEDV), and end systolic (LVESV) volumes. Pulse-wave doppler imaging was used to acquire isovolumetric relaxation (IVRT) and contraction (IVCT) times, and with aortic ejection time (ET), used to calculate Tei index (IVCT + IVRT/ET). Tissue doppler taken at the mitral annulus was used to describe LV filling pressure described by E’, E’/A’, and E/E’. An electrocardiogram (ECG) was obtained using telemetry in mice at baseline, 7 days, and 28 days post-MI while under anesthesia, as previously described [52].

### 4.6. LC-MS/MS

Tissue samples were ground using a mortar and pestle on dry ice pre-cooled with liquid nitrogen, and stored at −80 °C until processing. LC–MS/MS methods were used as previously described [39].

### 4.7. Protein Immunoblotting

Tissues were processed by subcellular fractionation by differential centrifugation to obtain mitochondrial, microsomal, and cytosolic fractions of LV tissue, as previously described [33]. Briefly, frozen LV tissues were ground and homogenized in fractionation buffer (sucrose 250 mM, TrisHCL 10 mM, EDTA 1 mM, sodium orthovanadate 1 mM, sodium fluoride 1 mM, aproptinin 10 μg/L, leupeptin 2 μg/L, pepstatin 100 μg/L). Homogenate was first centrifuged for 10 min at 700× *g* (4 °C). The supernatant was decanted and centrifuged for 20 min at 10,000× *g* (4 °C). The resulting pellet was used as a “crude” mitochondrial fraction, and supernatant was centrifuged for 1 h at 100,000× *g* (4 °C). The subsequent pellet was taken as the microsomal fraction, which contains endoplasmic reticulum (ER), and supernatant as cytosolic fractions. Both mitochondrial and microsomal pellets were resuspended in fractionation buffer and protein concentrations of all fractions were determined using a Bradford assay. Aliquots of protein (25 μg) were loaded onto SDS-polyacrylamide gels, run at 90 V, and transferred onto 0.2 µM PVDF membranes. Membranes were probed for expression of sEH (ELabscience, EAB-10489, Houston, TX, USA), mEH (sc135984) (Santa Cruz Biotechnology Inc., Dallas, TX, USA), CYP2J2 (ABS1605, MilliporeSigma, Oakville, ON, Canada), SDHa (ab5839s), MFN1 (ab104274), Citrate Synthase (ab129095), VDAC (ab14734), α-tubulin (ab4074), (Abcam, Toronto, ON, Canada, OPA1 (bd612606) (Becton Dickinson Canada Inc., Mississauga, ON, Canada), MFN2 (cs9482), COX IV (cs11967), and GAPDH (cs5174) (Cell signaling Technology, Inc., New England Biolabs, Ltd., Whitby, On, Canada). Quantitation of band intensity was done using Image J software (NIH, USA) to obtain relative protein expression. All treatment groups were normalized to the appropriate loading control lane.

### 4.8. Mitochondrial Enzyme Activities

Enzymatic activity of key electron transport enzymes was assessed in human and mouse LV tissue spectrophotometrically [53]. LV tissue was ground on dry ice with a mortar and pestle cooled with liquid nitrogen. Samples were homogenized in ice-cold muscle homogenization buffer (20 mM Tris, 40 mM KCl, 2 mM EGTA, pH = 7.4, with 50 mM sucrose added the day of homogenization) and spun at 600× *g* for 10 min at 4 °C to remove cellular debris. Supernatant was collected and used to assess the activity of NADH:ubiquinone oxidoreductase (complex I), succinate dehydrogenase (SDH, complex II), cytochrome C oxidase (COX IV, complex IV), and citrate synthase (CS). Activity was normalized to volume and protein concentration, following protein determination with standard Bradford assay.

Sirtuin 3 (SIRT3) activity was detected in the isolated mitochondrial fractions using a Sirtuin 3 fluorescent assay kit (BPS Bioscience, San Diego, CA, USA), according to the manufacturer’s instructions [15]. In this assay, mitochondrial fractions were first isolated from the sham control or the non-infarct regions of the MI hearts obtained from female WT, sEH null, or *t*AUCB treated mice. Mitochondria fractions were mixed with the specific HDAC fluorogenic substrate, bovine serum albumin, NAD^+^, and sirtuin assay buffer. The deacetylation process induced by SIRT3 in the sample sensitizes the HDAC substrate so that subsequent treatment with the SIRT3 assay developer produced a fluorescence product that was measured using a fluorescence plate reader at 350/460 nm excitation/emission wavelengths, and activity was expressed as U/μg protein.

### 4.9. Statistics

All statistics were done using Prism 8 software. For human tissue samples, analysis was done using one-way ANOVA with Tukey’s post-hoc test. For animal experiments, two-way ANOVA was carried out followed by Tukey’s post-hoc testing. In all cases significance was set at *p* < 0.05.

## Figures and Tables

**Figure 1 ijms-22-01691-f001:**
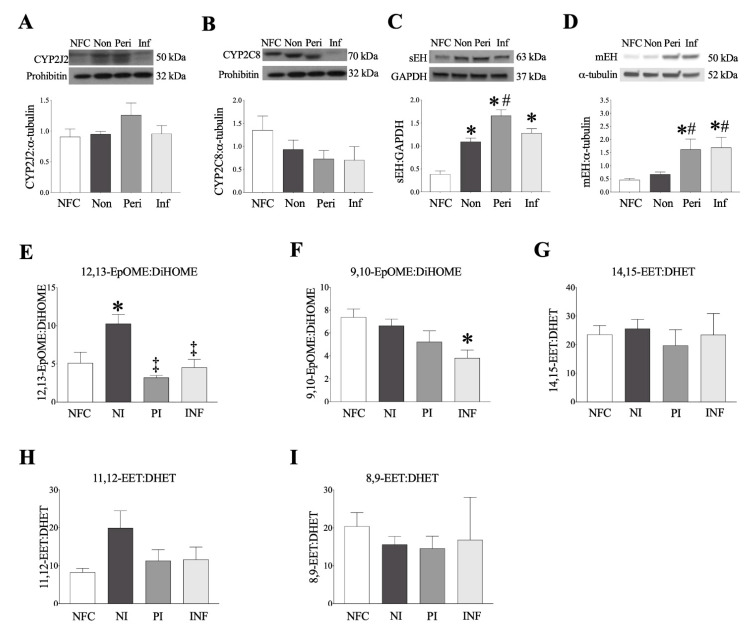
**Epoxide hydrolase, cytochrome P450 expression and oxylipid metabolites in human left ventricular tissue.** (**A**) CYP2J2, (**B**) CYP2C8, (**C**) soluble epoxide hydrolase (she) and (**D**) microsomal epoxide hydrolase (mEH) expression in non-failing control and myocardial infarction (MI) female hearts. CYP2J2 and CYP2C8 were normalized to prohibitin, sEH was normalized to GAPDH, and mEH normalized to α-tubulin. (**E**–**I**) Oxylipid metabolites in human left ventricle (LV) tissue (ng/g tissue) as measured by LC–MS/MS. (**E**) 12,13-EpOME:DiHOME ratios, (**F**) 9,10-EpOME:DiHOME ratios, (**G**) 14,15-EET:DHET ratios, (**H**) 11,12-EET:DHET ratios, (**I**) 8,9-EET:DHET ratios. Data are shown as mean ± SEM, N = 3–5. In all cases, *p* ≤ 0.05, * vs. control group; # vs. wild-type (WT) group; ‡ vs. non-infarct.

**Figure 2 ijms-22-01691-f002:**
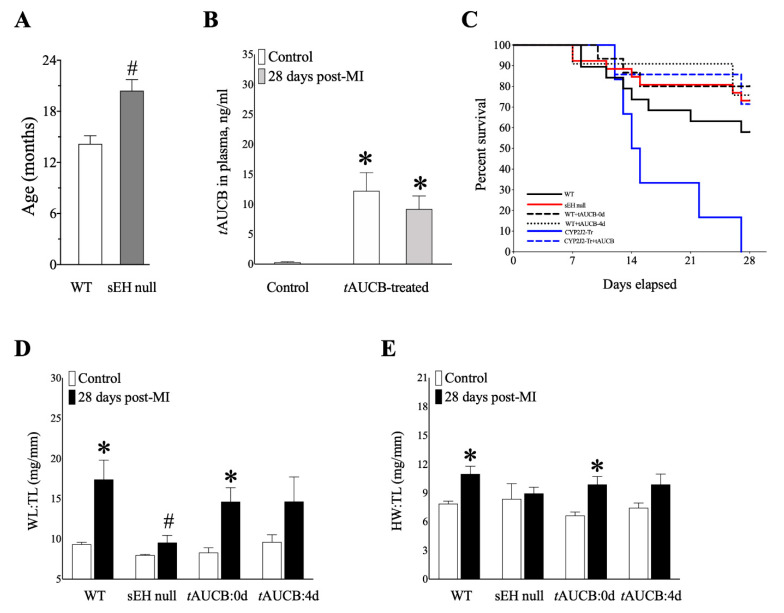
**Physiological parameters and survival for female mice at baseline and 28 days post-MI.** (**A**) Lifespan (months) for WT and sEH null female mice determined over natural aging. *n* = 40–46. (**B**) *t*AUCB levels (ng/mL) in plasma for WT vehicle-treated sham control mice and *t*AUCB-treated sham and 28 days post-MI females. (**C**) Percentage survival of female mice over 28 days of LAD ligation for WT (11/17, 58%) sEH null (19/26, 73%), WT+*t*AUCB:0d (12/15, 80%), WT+*t*AUCB:4d (5/6, 83%), CYP2J2-Tr (0/6, 0%), CYP2J2-Tr+tAUCB:0d (5/7, 71%). (**D**) Wet lung (WL) to tibia length (TL) (mg/mm) and (**E**) heart weight (HW) to TL (mg/mm) of WT, sEH null, tAUCB:0d- and tAUCB:4d-treated mice for control and at 28-days post-MI. Data are means ± SEM, *n* = 3–10. *p* ≤ 0.05; * vs. Control; # vs. WT group.

**Figure 3 ijms-22-01691-f003:**
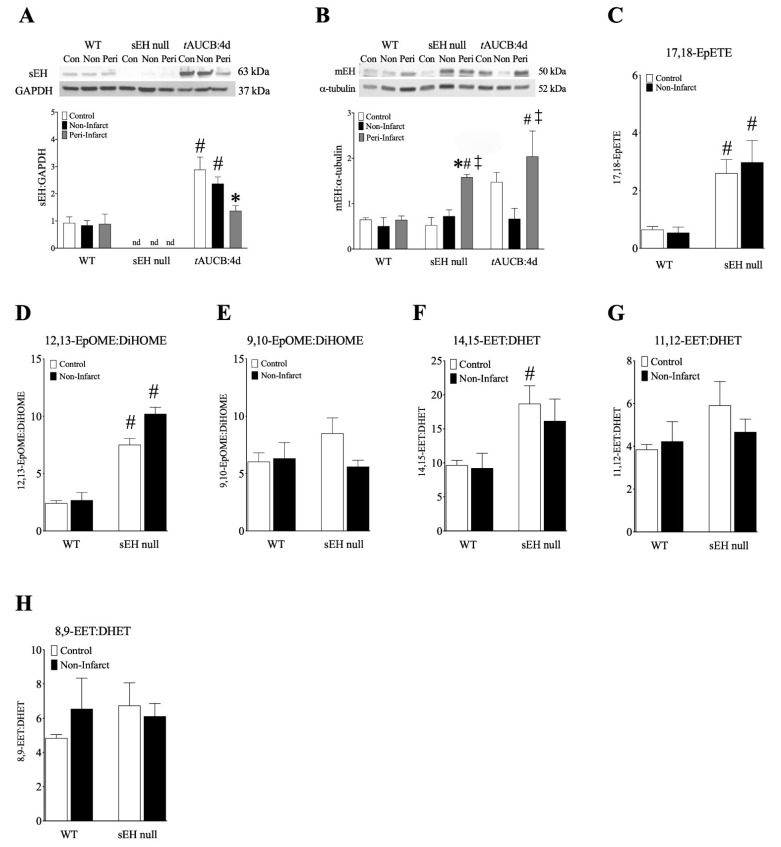
**Epoxide hydrolase expression and oxylipid metabolites (ng/g tissue) in female mouse left ventricular tissue.** (**A**) sEH expression normalized to GAPDH, and (**B**) mEH expression normalized to α-tubulin in control, non-infarct, and peri-infarct regions of female WT, sEH null, and WT+tAUCB:4d hearts at 28 days post-MI. (**C**) 17,18-EpETE, (**D**) 12,13-EpOME:DiHOME ratios, (**E**) 9,10-EpOME:DiHOME ratios, (**F**) 14,15-EET:DHET ratios, (**G**) 11,12-EET:DHET ratios, (**H**) 8,9-EET:DHET ratios. Data are mean ± SEM, *n* = 3–6. In all cases, *p* ≤ 0.05, * vs. control group; # vs. WT group; ‡ vs. non-infarct group.

**Figure 4 ijms-22-01691-f004:**
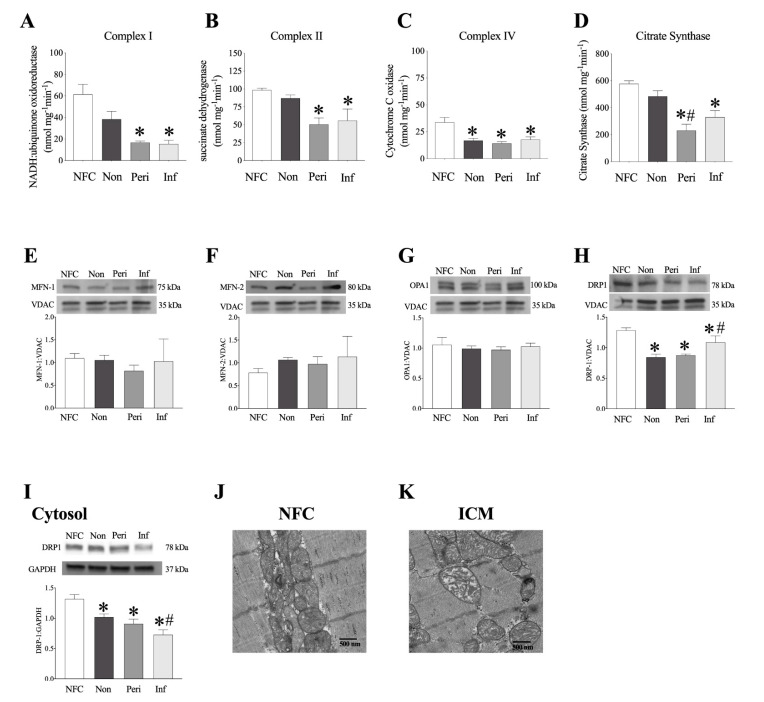
**Electron transport chain activity and mitochondrial protein expression in female human hearts.** (**A**–**D**) electron transport chain (ETC) complex activity (nmol mg^−1^min^−1^) in non-failing control and MI female hearts. (**A**) Complex I (NADH:ubiquinone oxidoreductase) activity, (**B**) Complex II (succinate dehydrogenase, SDH), (**C**) Complex IV (cytochrome C oxidase, COX IV), and (**D**) citrate synthase (CS) activity. Mitochondrial (**E**) MFN-1, (**F**) MFN-2, (**G**) OPA1, (**H**) DRP-1 expression and representative blots normalized to VDAC. (**I**) Cytosolic DRP-1 normalized to GAPDH. Data are represented as mean ± SEM, *n* = 3–5. *p* ≤ 0.05; * vs. NFC, # vs. non-infarct. Representative transmission electron micrograph images taken from female NFC (**J**) and ICM (**K**) human hearts. Magnification set at 10,000×.

**Figure 5 ijms-22-01691-f005:**
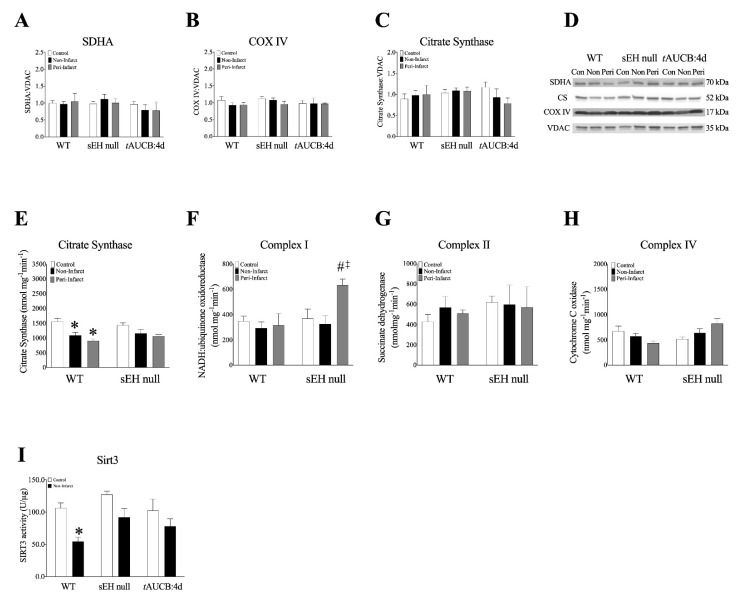
**Electron transport chain protein expression and activity in female mouse hearts.** (**A**–**D**) Protein expression of key mitochondrial ETC enzymes in mitochondrial fractions from female mouse left ventricular tissue. (**A**) Succinate dehydrogenase A (SDHA) expression, (**B**) Cytochrome C oxidase (COX IV) expression, (**C**) Citrate synthase (CS) expression, (**D**) representative blots. All proteins were normalized to VDAC. (**E**–**H**) ETC complex activity (nmol mg^−1^min^−1^) in WT and sEH null female left ventricular tissue. (**E**) citrate synthase (CS) activity, (**F**) Complex I (NADH:ubiquinone oxidoreductase) activity, (**G**) Complex II (succinate dehydrogenase, SDH), and (**H**) Complex IV (cytochrome C oxidase, COX IV). (**I**) Cardiac SIRT3 activity was determined in mitochondrial fractions in mice hearts using a SIRT3 fluorescent assay kit. Data are represented as mean ± SEM, N = 3–5. Data are represented as mean ± SEM, *n* = 3–5. *p* ≤ 0.05; * vs. Control, # vs. WT group, ‡ vs. non-infarct group.

**Figure 6 ijms-22-01691-f006:**
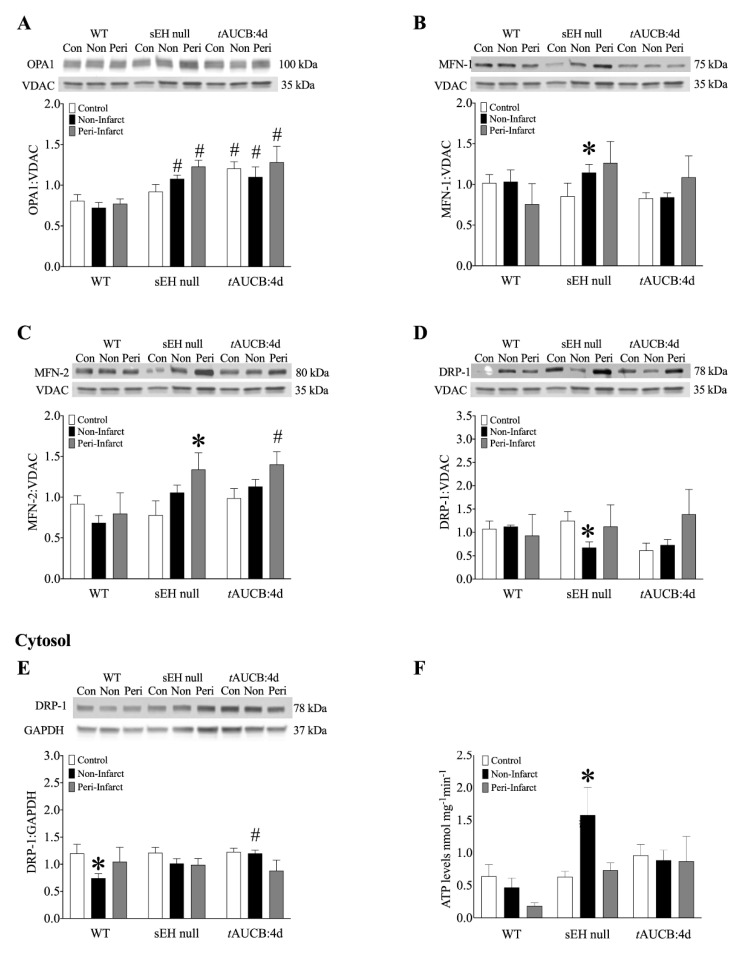
Protein expression in mitochondrial (**A**–**D**), and cytosolic (**E**) fractionates from female mouse hearts at 28 days post-MI. (**A**) OPA1, (**B**) MFN-1, (**C**) MFN-2, and (**D**) DRP-1 normalized to VDAC in isolated mitochondria in female WT, sEH null, and tAUCB:4d control, non-infarct and peri-infarct regions at 28 days post-MI. (**E**) Cytosolic DRP1 normalized to GAPDH, (**F**) ATP levels in nmol mg-1min-1 from WT, sEH null, and tAUCB:4d pre-treatment female mice in control and post-MI hearts. Data are means ± SEM, N = 3–6. *p* < 0.05; * vs. Control; # vs. WT group.

**Figure 7 ijms-22-01691-f007:**
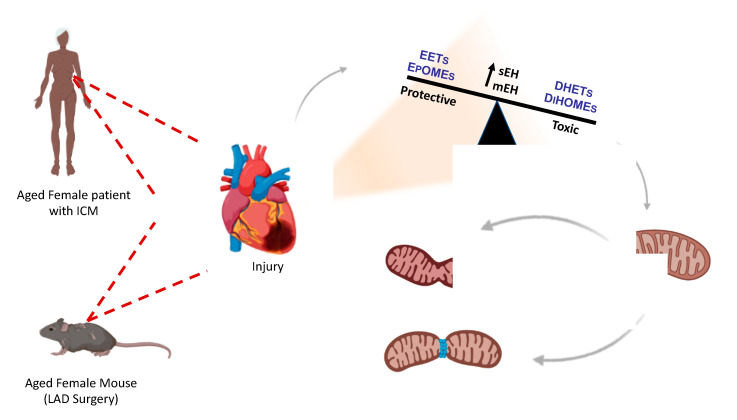
**A proposed mechanism of protection arising from sEH inhibition or deletion in female hearts.** Hearts where sEH activity is not inhibited are marked by increased mitochondrial dysfunction, manifesting with disrupted dynamics, impaired electrical signalling, and mechanical function at 28-days post-ischemia. This was associated with increased diol metabolite formation. In contrast, female hearts where sEH is genetically deleted or pharmacologically inhibited are marked by reduced mitochondrial dysfunction, cumulating in preserved ATP production and slowing the progression of cardiac dysfunction. Interestingly, pre-treatment with sEHi demonstrates a higher degree of cardioprotection, suggesting sEHi protection is time-dependent.

**Table 1 ijms-22-01691-t001:** Cardiac functional parameters in female mice at baseline, 7, and 28 days post-MI, measured by 2D echocardiography and electrocardiogram. Data are shown as mean ± SEM, *n* = 5–18. Data from two-way ANOVA with Tukey’s post hoc test. *p* ≤ 0.05, * vs. baseline; # vs. WT counterpart; ‡ vs. 7 days post-MI.

	WT	sEH null	WT+*t*AUCB:4d PreTx
ECHOCARDIOGRAPHY	Baseline	7d	28d	Baseline	7d	28d	Baseline	7d	28d
HR, beats/min	482 ± 9	469 ± 13	488 ± 11	499 ± 11	486 ± 11	512 ± 12	463 ± 13	476 ± 22	426 ± 7
*Wall measurements*									
Corrected LV mass, mg	111.4 ± 9.8	146.3 ± 12.3	149.4 ± 9.9 *	93.9 ± 3.2	114.7 ± 7.2	134.3 ± 12.4 *	94.80 ± 5.9	132.1 ± 13.5 *	119.8 ± 14.0
IVS-diastole, mm	0.89 ± 0.03	0.69 ± 0.10	0.60 ±0.06 *	0.87 ± 0.02	0.71 ± 0.05	0.70 ± 0.07	0.78 ± 0.04	0.74 ± 0.09	0.65 ± 0.17
IVS-systole, mm	1.32 ± 0.06	0.91 ± 0.14 *	0.76 ± 0.07 *	1.32 ± 0.04	0.91 ± 0.07 *	0.98 ± 0.10 *	1.18 ± 0.07	1.42 ± 0.20	0.78 ± 0.21
LVPW-diastole, mm	0.90 ± 0.04	0.94 ± 0.07	0.82 ± 0.06	0.85 ± 0.02	0.78 ± 0.06	0.81 ± 0.05	0.81 ± 0.04	0.90 ± 0.08	0.84 ± 0.05
LVPW-systole, mm	1.29 ± 0.05	1.01 ± 0.09 *	0.99 ± 0.10 *	1.25 ± 0.04	0.97 ± 0.08 *	1.04 ± 0.07	1.25 ± 0.07	1.20 ± 0.11	1.14 ± 0.13
LVID-diastole, mm	3.97 ± 0.13	5.10 ± 0.17 *	5.69 ± 0.16 *	3.80 ± 0.09	4.77 ± 0.21 *	5.12 ± 0.14 *	4.04 ± 0.10	4.78 ± 0.24 *	4.87 ± 0.22
LVID-systole, mm	2.66 ± 0.13	4.43 ± 0.28 *	5.04 ± 0.21 *	2.43 ± 0.10	3.97 ± 0.26 *	4.34 ± 0.20 *	2.65 ± 0.11	3.76 ± 0.36 *	3.92 ± 0.29*
*Cardiac Function, Simpsons*									
EF, %	59.17 ± 1.58	27.29 ± 1.38 *	23.13 ± 3.76 *	65.29 ± 1.53	37.05 ± 2.44 *#	30.25 ± 3.03 *	61.00 ± 2.99	41.91 ± 5.00 *#	33.53 ± 5.15 *
FAC, %	48.86 ± 2.38	17.96 ± 2.87 *	16.63± 3.04 *	55.92 ± 2.44	29.34 ± 3.10 *#	22.94 ± 2.06 *	53.63 ± 3.27	31.75 ± 5.61 *#	29.55 ± 2.61 *
LVEDV, µL	75.76 ± 4.20	140.86 ± 16.13 *	179.78 ± 19.63 *	66.74 ± 2.85	116.31 ± 10.40 *	138.18 ± 11.23 *	64.65 ± 6.15	108.39 ± 17.54 *	114.53 ± 9.42 *
LVESV, µL	31.37 ± 2.56	103.20 ± 12.78 *	143.07 ± 20.50 *	23.44 ± 1.72	75.87 ± 9.19 *	99.70 ± 11.31 *#	25.50 ± 4.08	71.47 ± 16.23 *	69.10 ± 10.20 *
CO, mL/min	21.31 ± 1.00	17.51 ± 1.89	16.86 ± 1.64	21.96 ± 0.84	20.32 ± 1.17	19.57 ± 1.41	18.18 ± 1.14	17.91 ± 2.25	22.82 ± 3.01
SV, µL	44.39 ± 2.09	37.66 ± 3.67	35.97 ± 3.41	43.59 ± 1.71	40.44 ± 1.95	38.48 ± 2.97	39.15 ± 2.71	36.92 ± 3.87	45.44 ± 5.02
*Doppler Imaging*									
IVRT, ms	16.02 ± 0.78	18.74 ± 2.07	19.95 ± 1.94	15.27 ± 0.94	16.77 ± 0.91	25.67 ± 4.57 *‡	16.27 ± 0.79	19.91 ± 1.09	21.87 ± 2.81
IVCT, ms	14.96 ± 1.16	19.64 ± 2.75	31.28 ± 7.58 *‡	11.12 ± 0.64	20.99 ± 3.04 *	18.99 ± 3.50 #	15.22 ± 0.96	23.17 ± 2.73	26.31 ± 0.78
ET, ms	40.77 ± 1.45	36.51 ± 2.35	39.22 ± 2.37	39.71 ± 0.89	38.47 ± 1.57	35.60 ± 2.08	46.55 ± 1.14 #	42.79 ± 0.96 #	51.96 ± 6.38 #
Tei index	0.76 ± 0.03	1.09 ± 0.08 *	1.25 ± 0.11 *	0.67 ± 0.03	1.00 ± 0.13 *	1.24 ± 0.20 *	0.68 ± 0.01	1.01 ± 0.08 *	0.94 ± 0.04
E’	23.71 ± 1.48	22.24 ± 4.50	13.61 ± 3.01 *	25.85 ± 1.50	18.51 ± 1.74 *	16.54 ± 2.47 *	27.72 ± 1.96	19.32 ± 1.26 *	25.38 ± 10.87
E’/A’	1.15 ± 0.05	1.28 ± 0.25	1.22 ± 0.32	1.32 ± 0.13	1.25 ± 0.11	0.98 ± 0.10	1.35 ± 0.08	1.68 ± 0.32	0.96 ± 0.14
E/E’	22.53 ± 1.23	32.16 ± 6.37	45.23 ± 7.72 *	23.05 ± 1.36	34.47 ± 4.93 *	41.50 ± 4.69 *	21.26 ± 1.84	29.83 ± 4.61	33.64 ± 12.42
ELECTROCARDIOGRAM									
HR, beats/min	471 ± 14	488 ± 11	523 ± 5	505 ± 21	475 ± 31	529 ± 14			
RR, ms	128.1 ± 3.8	123.3 ± 2.9	114.8 ± 1.0	121.6 ± 4.3	129.4 ± 8.2	112.7 ± 4.0			
QRS, ms	12.0 ± 0.3	10.7 ± 0.2	15.1 ± 1.7 *‡	11.7 ± 0.2	10.3 ± 0.4	10.9 ± 1.6 #			
PR, ms	43.3 ± 1.5	40.8 ± 1.6	51.8 ± 3.2 *‡	42.1 ± 1.0	42.0 ± 0.3	44.6 ± 1.4 #			
QTcF, ms	45.7 ± 0.8	74.0 ± 2.2 *	65.8 ± 7.6 *	45.1 ± 1.0	70.2 ± 5.1 *	58.5 ± 1.1

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
