# Peer review of "Soluble Epoxide Hydrolase in Aged Female Mice and Human Explanted Hearts Following Ischemic Injury"

_ijms, 2021, doi:10.3390/ijms22041691_

Round 1

Reviewer 1 Report

The authors have addressed all my comments in an appropriate manner. I still think, that this is a good paper.

Reviewer 2 Report

Dear authors, thank you for the revision.

This manuscript is a resubmission of an earlier submission. The following is a list of the peer review reports and author responses from that submission.

Round 1

Reviewer 1 Report

According to the manuscript, the aim of the study was to “… characterize oxylipid metabolism in the left ventricle (LV) following ischemic injury in females”.

The aim of the study is interesting, however, if one wants to reveal any sex-dependent differences, the comparison with other sex (males) and the exact characteristics of the conditions (i.e. hormonal status) is needed.

With that, there is no such data in the manuscript.

If the study was aimed to reveal the age-dependent differences (the authors apply to perimenopausal and menopausal age), the hormonal changes should be studied as well. Thus, the age of the animals used in the study was 13-16 months. Around 9–12 months of age, rats, and mice typically experience irregular estrous cycles and do not become acyclic or have low estrogen (menopause) until 18–24 mo. of age (variable between mouse strains). What was the hormonal status of animals? Did the authors conduct the determination of the stages of the estrous cycle in animals? How the age of the animals corresponded to the age of humans, whose samples were investigated? With the absence of these data there is no reason to say that the authors “use a clinically relevant animal model to demonstrate aged female mice”

During the study, human LV specimens were procured from female patients with ischemic cardiomyopathy (ICM) or non-failing controls (NFC). According to Supplementary files, the age and BMI of on-failing control (NFC) were 48 years and 24.4 respectively; while for the patient with ischemic cardiomyopathy (ICM)—58 years and 26.5.  With that, in females, menopause, the cessation of menstrual cycling, is associated with an increase in risk for several diseases such as cardiovascular disease. What was the status of female patients in the study? The age of patients varies between the groups.  There is no explanatory data in the manuscript.

Taken together, the author’s conclusions that “data presented here suggest inhibiting sEH post-ischemia will have perior outcomes in females” are not justified due to the methodological errors in the study.

Minor concerns. English language quality must be improved. I suggest the authors get editing help from someone with full professional proficiency in English.

Reviewer 2 Report

The study "Soluble Epoxide Hydrolase in aged female mice and human explanted hearts following ischmeic injury" by Jamieson et al. is a great story, that investigates extensively why female hearts are at more risk during myocardia infaction. The authors show a wide display of mitochondrial regulation mechanisms, that may be responsible for this already well-known phenomenon. This is a great, comprehensive paper that answers al ot of questions in this field - great- please publish.It is fun to read and follow.

I have not detected any major or minor flaws, therefor I have only a few micro points to adress. I need not to see this for a Re-Review, the editor may decide, but this is close to perfect.

My Micro points:

  1. page 2, line 16-19: The authors may add one or two sentences, what is different in female mitochondria.
  2. All figures: I belong to the individuals, tht love to print out a paper and look at the figures. However, on a print out all these figures are on a miniature scale. Since this will be an online publication, the figures couls be a little bit bigger, that they are visisble on paper without using a 5x magnifying glass.
  3. All figures: The figures are nice, but too busy. Is it really necesary to display every value as a spot in those colums. This makes the figures somehow spotty and confusing. The passionate reader should be able to judge SEM bars with a littel star on it in the case of statistical significance.
  4. Fig 3 A and B. The third column of each set ("post-infarct") should be the same gray scale as the figure legend. This is not the case (or it is not visible).
  5. Page 8 line 7 Please show some (e.g.the most impressive with respect on the question) of the original Echograms in supplemental videos. There is supplemental material mentioned, but I did not get acccess to it. It may be somethings else.
  6. Table 1 could be a little larger font scale for easier reading. There should be space enough on this page
  7. Figure 4 J and K: Can you make this bigger and enhance contrast?
  8. Fig 5 and 6 are further perfections of miniaturism!
  9. nice sketch for the proposed mechanism!
  10. Methods: statistical significane is usually described as < 0.05; not just <0.05
  11. I like, that for the most experimental sets only 3-6 animals were used to get a clear result. This is a good ethical point instead of sacrificing hundreds of animals to answer just one experrimental question.

Round 2

Reviewer 1 Report

Dear authors, unfortunately, despite the work done, my opinion has not changed. It is not reasonable to draw such conclusions based on limited information.
At the same time, the results obtained can be published as a pilot study, discussing all the limitations and future perspectives in a special section.